# Escaping the Reality of the Pandemic: The Role of Hopelessness and Dissociation in COVID-19 Denialism

**DOI:** 10.3390/jpm12081302

**Published:** 2022-08-10

**Authors:** Chiara Ciacchella, Giorgio Veneziani, Claudio Bagni, Virginia Campedelli, Antonio Del Casale, Carlo Lai

**Affiliations:** 1Department of Dynamic and Clinical Psychology, and Health Studies, Sapienza University of Rome, 00185 Rome, Italy; 2Unit of Psychiatry, Sant’ Andrea University Hospital, 00189 Rome, Italy

**Keywords:** denialism, coronavirus, dissociation, sense of community, hopelessness

## Abstract

Background: Denialism of coronavirus disease 2019 (COVID-19) severely affected governments’ attempts to contain the spread of the virus. Indeed, groups of deniers showed scepticism and misinformation toward the causes of the virus, leading to less adherence to official guidelines and vaccination campaigns. The present study aimed to investigate the sociodemographic and psychological factors associated with COVID-19 denialism, expressed in the forms of scepticism, nonadherence to guidelines, and negative attitudes toward vaccination. Methods: Four hundred and sixty-one volunteers completed an online survey composed of the Beck Hopelessness Scale, the Dissociative Experiences Scale-II, the Sense of Community Index, and a questionnaire about COVID-19 denialism. Results: The multiple regression analyses showed that higher age and a lower level of education were positive predictors of COVID-19 denialism. Furthermore, the structural equation model showed that hopelessness positively predicted dissociation and negatively predicted the sense of community. In turn, only dissociation was found to positively predict COVID-19 denialism. Conclusions: The findings of the present study suggested that hopelessness could exacerbate a defensive dissociative response that could be associated with greater COVID-19 denialism. Moreover, older and less educated people showed a greater propensity to engage in COVID-19 denialism.

## 1. Introduction

Coronavirus disease 2019 (COVID-19) continues to present enormous health, social, and economic challenges for communities around the world. The unexpected epidemiological nature of COVID-19 has required both individual and community efforts and the investment of many resources to stem the pandemic. Governments have taken important preventive measures to limit the spread of infection, such as the mandatory use of masks, social distancing, and quarantines, which have led to profound changes in daily life, with serious impacts on the mental health of the population [1,2]. At the same time, medical research has been promoted and funded to identify effective strategies to treat and prevent the virus. Research on specific antiviral drugs is still ongoing, and to date, vaccination is the safest and most effective way to prevent the impact of COVID-19 and its future variants [3,4]. Countries that have conducted large-scale vaccination have been able to loosen strict preventive measures, alleviating their negative psychological impact on the population [3,4].

Although scientific evidence has shown that the strategies adopted (preventive measures and vaccination) were effective in reducing the spread of the virus, many people were sceptical about the truthfulness of the official data. Since the beginning of the pandemic, groups of deniers and anti-vaccination movements have spread all over the world [5,6]. People who joined these groups showed disbelief in COVID-19, distrusted their governments, and started to not adhere to preventive measures and official guidelines, putting themselves and others at risk [5,7,8,9,10]. These groups also spread misinformation about vaccine safety, causing a worrying reduction in vaccination rates [11,12].

Because of the need to overcome the pandemic, it seems extremely relevant to understand the factors that may influence denialism and adherence to anti-vaccination movements. An interesting perspective could be to consider denialism as a strategy to cope with the unprecedented uncertainty of the pandemic. People most vulnerable to COVID-19, such as older adults, may cope with the high level of stress and hopelessness by denying the seriousness of the pandemic situation and adopting conspiratorial beliefs [13]. A similar pattern seems to be applicable to less educated people who tend to ignore scientific methods and embrace pseudo-scientific theories instead [13,14,15]. From this perspective, the overwhelming stress and sense of hopelessness associated with the outbreak could be temporarily alleviated through immature defensive mechanisms, such as dissociation, which would reduce awareness of intolerable information [16,17]. Some research has indicated that belief in conspiracy theories could be increased by dissociation [18].

In contrast, a more adaptive strategy for coping with the sense of hopelessness and uncertainty associated with COVID-19 might be to seek support from community members. Indeed, a sense of community, characterised by feelings of belongingness, attachment to a place, and emotional connection with other members of the community, seems to be an important factor in fostering resilience and positive well-being [19,20,21]. In this regard, previous studies have reported that people who trust their communities show lower levels of hopelessness about the future [22,23], as well as a lower tendency to endorse theories of denialism [24]. In addition, feeling responsible for other community members may promote greater adherence to official guidelines and vaccination [25,26,27].

Considering the urgent need to promote adherence to government guidelines and to achieve global vaccination coverage, the different psychological factors related to COVID-19 denialism should be explored. The present study aims to investigate the sociodemographic and psychological factors associated with COVID-19 denialism, as measured by scepticism about the scientific explanations, nonadherence to the guidelines, and a negative attitude towards vaccination. The first hypothesis was that a sense of hopelessness would be associated with a greater dissociative response and a lesser sense of community, which in turn would be associated with greater COVID-19 denialism. The second hypothesis was that higher age and a low level of education would predict COVID-19 denialism.

## 2. Materials and Methods

The study was approved by the Institutional Ethics Committee of Department of Dynamic and Clinical Psychology and Health Studies, Sapienza (11/06/21, No 0000797). The present study complied in accordance with the World Medical Association Declaration of Helsinki (1964). Four hundred and sixty-one volunteers signed the informed consent and completed the online survey from 27 July 2021 to 3 December 2021. The inclusion criteria were that participants were residents of Italy and that they were able to read and understand Italian. The exclusion criterion was participants aged <18 years.

### 2.1. Materials

Through the Google Forms platform, an online survey was composed ad hoc to obtain information about participants’ age, gender, country of residence, and education level. Subjects were invited to participate in the study through different platforms and social networks (Instagram, Telegram, WhatsApp, and Facebook). The psychological evaluation was conducted through the administration of the following self-report questionnaire.

The Beck Hopelessness Scale (BHS) [28,29] was used to determine the negative expectations of the individual for the future. The BHS consists of 20 items, and each question is scored between 0 and 1. High scores indicate higher levels of hopelessness.

The Dissociative Experience Scale-II (DES-II) [30,31] is a self-assessment instrument composed of 28 items that measure the level and type of dissociative experience. This tool uses an 11-point percentage scale (0–100%) on which the participant rates their experience. The total score is between 0 and 100; scores below 30 are frequently found in healthy controls and in psychiatric patients in general, and scores above 30 are generally associated with the presence of significant dissociative symptoms [30].

The Sense of Community Index 2 (SCI-II) [32] was used to measure participants’ sense of community related to their own city. The SCI-II is a tool for measuring a sense of community where participants can express agreement with the proposed sentences by scoring on a 4-point Likert scale, ranging from minimal agreement (0 = not at all) to maximum agreement (3 = completely). The scale is based on a theory by McMillan and Chavis [33] that suggests a sense of community is expressed in the perception of four aspects, measured by four subscales: (1) Reinforcement of Needs, which corresponds to the idea that common needs, goals, beliefs and values provide the integrative force for a cohesive community that can meet both collective and individual needs; (2) Membership, which corresponds to feelings of belonging and emotional security arising from being part of a defined community; (3) Influence, which corresponds to the members’ feelings of control and influence over the community; and (4) Shared Emotional Connection, which corresponds to the bonds developed over time through positive interaction with other community members [32].

Finally, COVID-19 denialism was assessed using the following three self-administered items, rated on a Likert scale that ranged from 0 (strongly disagree) to 7 (strongly agree): (1) scepticism, “I feel scepticism about the official explanations for the causes of the virus”; (2) nonadherence to the guidelines, “I refuse to adhere to the guidelines imposed by the government, as I believe that the pandemic is not a real problem”; and (3) a negative attitude towards vaccination, “I am in favour of the use of vaccines against the COVID-19 virus”. To obtain a measure of the negative attitude towards vaccination, this last item was reverse-scored.

### 2.2. Statistical Analysis

Correlation analyses (Pearson’s *r*) were performed between the sociodemographic variables (gender, age, and level of education), dissociation (DES-II total score), the sense of community (the SCI-II subscales of Reinforcement of Needs, Membership, Influence, and Shared Emotional Connection), and COVID-19 denialism items (scepticism, nonadherence to the guidelines, and negative attitude toward vaccination). Correlations (Pearson’s *r*) were also computed between hopelessness (BHS total score), dissociation (DES-II total score), and sense of community (SCI-II subscales).

To test the first hypothesis, multiple regression models were performed on each COVID-19 denialism item (scepticism, nonadherence to the guidelines, and negative attitude toward vaccination) including the sociodemographic variables (age and level of education) as predictors. The correlation and regression analyses were performed using Statistica 8.0 (StatSoftInc., Tulsa, OK.).

A structural equation model (SEM) was designed to test the second hypothesis using the SEM package of the JASP software (2021) for Mac. The SEM, as estimated using the maximum likelihood method, and the model fit were assessed using the following indices: The comparative fit index (CFI), Bentler–Bonett nonnormed fit index (NNFI), root-mean-square error of approximation (RMSEA), and the standardised root mean square residual (SRMR). The following criteria for a good fit were considered: ≥0.97 for CFI and NNFI, ≤0.05 for RMSEA, and ≤0.05 for SRMR [34].

## 3. Results

A total of 461 participants (379 women and 82 men) were enrolled in the present study. The mean age was 29.1 years (*SD* = 10.3), and the mean level of education was 14.9 years of schooling (*SD* = 2.7).

Table 1 showed the correlations (Pearson’s *r*) computed between the sociodemographic variables (gender, age, and level of education), dissociation (DES-II total score), sense of community (Reinforcement of Needs, Membership, Influence, and Shared Emotional Connection), and COVID-19 denialism (scepticism, nonadherence to the guidelines, and negative attitude toward vaccination). Age was positively correlated with scepticism, nonadherence to the guidelines, and a negative attitude towards vaccination. Education level was negatively correlated with scepticism, nonadherence to the guidelines, and a negative attitude towards vaccination. Dissociation was positively correlated with scepticism and nonadherence to the guidelines. Regarding the sense of community, the Reinforcement of Needs subscale was negatively correlated with a negative attitude towards vaccination; the Membership subscale was positively correlated with nonadherence to the guidelines; the Influence subscale was positively correlated with nonadherence to the guidelines; and the Shared Emotional Connection subscale was negatively correlated with a negative attitude toward vaccination.

In Table 2 the correlations (Pearson’s *r*) performed between hopelessness (BHS total score) and dissociation (DES-II total score) and sense of community (Reinforcement of Needs, Membership, Influence, and Shared Emotional Connection) are shown. Hopelessness was positively correlated with dissociation and negatively correlated with sense of community (all the SCI-II subscales).

The multiple regression analyses performed with age and level of education as predictors of COVID-19 denialism (scepticism, nonadherence to the guidelines, and negative attitude toward vaccination), all showed significant models (Table 3). The significant models showed that age was a positive predictor, and level of education was a negative predictor of scepticism, nonadherence to the guidelines, and negative attitude toward vaccination (Table 3).

The goodness of fit of the structural equation models was tested on the whole sample. To create the “sense of community” latent variable, the scores obtained on the four SCI-II subscales (Reinforcement of Needs, Membership, Influence, and Shared Emotional Connection) were used. In addition, the covariances between the manifest variables converging toward the same latent variable (sense of community) were estimated, maintaining coherence with the conceptual model. To create the “COVID-19 denialism” latent variable, the scores of the items related to scepticism, nonadherence to the guidelines, and negative attitudes toward vaccination were used. The model was acceptable, and the fit indices approached the thresholds for a good fit [34]: χ^2^ = 65.6, df = 19, *p* < 0.001; CFI = 0.97; NNFI = 0.94; RMSEA = 0.07, 95% confidence interval [0.054, 0.093]; SRMR *=* 0.045. The model’s parameter estimates are reported in Figure 1. All the manifest variables loaded significantly on the respective latent variables (Figure 1). Hopelessness (BHS total score) was found to positively predict dissociation (DES total score) and negatively predict sense of community (Figure 1). Furthermore, dissociation positively predicted COVID-19 denialism, and sense of community was not found to be a significant predictor of it (Figure 1).

## 4. Discussion

The main finding of the present study was that a feeling of hopelessness about the future was related to a greater dissociative response, which in turn was associated with greater COVID-19 denialism. Specifically, levels of dissociation, predicted by hopelessness for the future, were positively associated with higher scepticism of the official information about COVID-19, reduced adherence to the norms, and a greater negative attitude toward COVID-19 vaccination. The global crisis due to COVID-19 has exacerbated feelings of uncertainty, fear and helplessness, which increased people’s loss of hope for the future [35,36]. It has previously been reported that a greater degree of uncertainty about the future can cause a defensive dissociative response [37,38] and, in the current context, denying the pandemic reality could be a way to detach oneself from an overwhelmingly emotional state. Instead of dealing with the lack of positive prospects, people could seek alternative explanations to help them accept the unpleasant reality.

Indeed, people who embrace denialist theories in response to the pandemic experience usually believe in global control mechanisms [39], according to which a small group of elites successfully pull the strings of complex processes, conspiring against ordinary people [39]. Faced with this alternative reality, the only possible solution is to oppose such a system, under the false impression that one is acting against these negative events. This kind of idea could explain the low adherence to guidelines, because these norms could be seen by COVID-19 deniers as something unnecessary, or even unfair, instead of being useful precautions to protect their own health and that of others.

It is interesting that, although the sense of community was negatively predicted by levels of hopelessness, it did not appear to be associated with COVID-19 denialism. Conversely, previous studies have reported that a feeling of belongingness and attachment to a place, accompanied by a sense of connection with other members of the community, could promote adherence to norms and prosocial behaviours [27,40,41,42,43]. Although the concept of a sense of community is generally analysed by referring to a geographically defined community, it is possible that a sense of belonging and a connection to one’s city are not sufficient to explain the complex phenomenon of denialism. It would be interesting to investigate whether feeling part of “relational” communities (e.g., professional, spiritual) [33] could be a protective factor against denialism.

A further important finding of this study was that higher scepticism of the official information about COVID-19 reduced adherence to the guidelines, and a greater negative propensity toward COVID-19 vaccination was positively predicted by age and negatively predicted by education level. It would therefore seem that older people and less educated one people have a greater propensity to engage in denialist attitudes [44,45]. Considering that older adults comprise one of the most fragile groups, it can be assumed that denying the evidence about COVID-19 and its consequences help them to avoid facing the fears that the pandemic crisis has unleashed [13,46]. Moreover, a lower education level was related to endorsement of COVID-19 denialism, which is consistent with previous studies that have found higher education levels to be negatively correlated with epistemically suspect beliefs [47,48,49,50,51,52].

Moreover, a low level of education appears to be associated with low health literacy [53], which corresponds to people’s ability to obtain, process, and understand health-related information to make appropriate decisions (Institute of Medicine, 2004). In this regard, a recent study reported a negative association between health literacy and belief in conspiracy theories related to COVID-19 [54].

Although these are interesting findings, the present study has some limitations. Despite the adequate sample size, most of the participants were women. Therefore, subsequent studies should recruit a more balanced gender distribution. In addition, the participants in the present study were not differentiated on the basis of vaccination status. Indeed, the aim of the study was to highlight factors associated with a negative attitude toward the COVID-19 vaccine and not about how these affected the actual vaccination rate. An interesting future project could investigate whether a greater dissociative response could be found specifically in nonvaccinated individuals. In addition, the use of self-report questionnaires may have resulted in the presence of desirability bias in responses. Future studies should include more objective measurement tools to detect the variables of interest. Finally, the study’s cross-sectional nature does not allow one to draw conclusions about causation.

## 5. Conclusions

The results of the present study suggested that denialism could be considered a strategy to cope with the negative impact that the pandemic has had. In this regard, interventions that promote more functional defensive mechanisms against the hopelessness about the future could reduce dissociation, positively affecting the adherence to, and trust in, official guidelines. Furthermore, governments and the healthcare system should implement communicative strategies aimed to reach older people and those with less education by promoting more informed and accessible health-related information. This strategy would enable the implementation of patient-centred interventions with a collaborative approach to care.

## Figures and Tables

**Figure 1 jpm-12-01302-f001:**
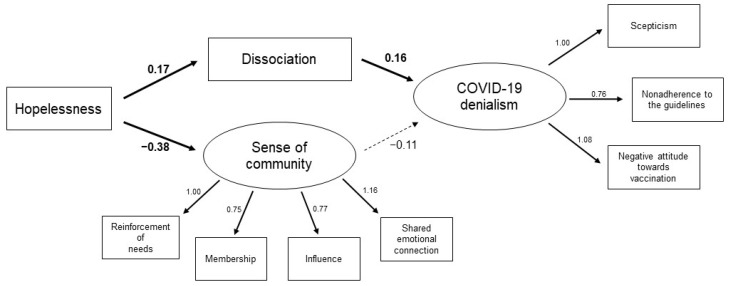
Estimate structural equation model with hopelessness as predictor of dissociation and sense of community, and dissociation as predictor of COVID-19 denialism (*n* = 461). The solid lines represent statistically significant paths (*p* < 0.05), and the dashed lines represent not significant paths.

**Table 1 jpm-12-01302-t001:** Correlations (Pearson’s *r*) computed between the sociodemographic variables (gender, age, and level of education), dissociation (Dissociative Experiences Scale II (DES-II) total score), sense of community (sense of community index II (SCI-II) subscales: Reinforcement of Needs, Membership, Influence, and Shared Emotional Connection), and COVID-19 denialism (scepticism, nonadherence to the guidelines, and negative attitude toward vaccination) (*n* = 461).

	COVID-19 Denialism		
	Scepticism	Nonadherence to the Guidelines	Negative Attitude toward Vaccination
Gender	*r =* 0.0799	*r =* −0.0463	*r =* 0.0177
Age	*r =* 0.1843 ***	*r =* 0.2747 ***	*r =* 0.2611 ***
Level of education	*r =* −0.1278 **	*r =* −0.1228 **	*r =* −0.1950 ***
Dissociation (DES-II total score)	*r =* 0.2563 ***	*r =* 0.1138 *	*r =* 0.0892
Sense of community (SCI-II Reinforcement of Needs)	*r =* −0.0821	*r =* −0.0118	*r =* −0.0980 *
Sense of community (SCI-II Membership)	*r =* 0.0001	*r =* 0.1005 *	*r =* −0.0365
Sense of community (SCI-II Influence)	*r =* −0.0363	*r =* 0.0933 *	*r =* −0.0302
Sense of community (SCI-II Shared Emotional Connection)	*r =* −0.0559	*r =* −0.0007	*r =* −0.1035 *

Note. * = *p* < 0.050; ** = *p* < 0.010; *** = *p* < 0.001

**Table 2 jpm-12-01302-t002:** Correlations (Pearson’s r) performed between hopelessness (Beck Hopelessness Scale (BHS) total score), dissociation (Dissociative Experiences Scale II (DES-II) total score), and sense of community (sense of community index II (SCI-II) subscales; Reinforcement of Needs, Membership, Influence, and Shared Emotional Connection) (*n* = 461).

	Dissociation	Sense of Community			
	DES-II Total Score	SCI-II Reinforcement of Needs	SCI-II Membership	SCI-II Influence	SCI-II Shared Emotional Connection
Hopelessness (BHS total)	r = 0.1745 ***	*r =* −0.3065 ***	*r =* −0.2287 ***	*r =* −0.1999 ***	*R =* −0.2870 ***

Note. *** = *p* < 0.001

**Table 3 jpm-12-01302-t003:** Multiple regression analyses performed with age and level of education as predictors of COVID-19 denialism (scepticism, nonadherence to the guidelines, and negative attitude toward vaccination) (*n* = 461).

Dependent Variable	Predictors	Beta	Std.Err. of Beta	B	Std.Err. of B	t (458)	*p*-Level
Scepticism (1)	Age	0.20	0.05	0.04	0.01	4.40	<0.001
	Level of education	−0.15	0.05	−0.12	0.04	−3.28	0.001
Nonadherence to the guidelines (2)	Age	0.29	0.04	0.03	0.01	6.54	<0.001
	Level of education	−0.15	0.04	−0.07	0.02	−3.47	<0.001
Negative attitude toward vaccination (3)	Age	0.29	0.04	0.05	0.01	6.48	<0.001
	Level of education	−0.23	0.04	−0.15	0.03	−5.13	<0.001
Models’ summaries (1) *R =* 0.24; R^2^ = 0.06; Adjusted R^2^ = 0.05; F(2,458) = 13.6; *p* < 0.001; Std.Error of estimate = 2.21 (2) *R =* 0.31; R^2^ = 0.09; Adjusted R^2^ = 0.09; F(2,458) = 25.2; *p* < 0.001; Std.Error of estimate = 1.15 (3) *R =* 0.34; R^2^ = 0.12; Adjusted R^2^ = 0.12; F(2,458) = 30.9; *p* < 0.001; Std.Error of estimate = 1.71							

## Data Availability

The data that support the findings of this study are available from the corresponding author, C.L., upon reasonable request.

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
