# Peer review of "Escaping the Reality of the Pandemic: The Role of Hopelessness and Dissociation in COVID-19 Denialism"

_jpm, 2022, doi:10.3390/jpm12081302_

Round 1
Reviewer 1 Report
The paper has a clear objective, the methods used are appropriate, and the conclusion can largely match the objective. The paper is appropriate for the Journal of Personalized Medicine. The writing and structure of the paper needs extensive revision. I suggest a major revision.
1. Line 32, use Covid-19, delete Coronavirus Disease 2019, as it as appeared in line 29.
2. Page three is a repetition of page two.
3. Line 159, the 2nd citation style appears.
4. line 184, "Through Correlations (Pearson’s r) were performed ", the english is problematic.
5. Line 209, english is problematic.
6. Table 1, the table is hard to read, use * rather than exact p value.
7. Table 2, the table is hard to read, use * rather than exact p value.
8. Table 3, the table is hard to read, move the R2, Adj R2 to the below, and make the table cleaner.
Author Response
Reviewer1:
The paper has a clear objective, the methods used are appropriate, and the conclusion can largely match the objective. The paper is appropriate for the Journal of Personalized Medicine. The writing and structure of the paper needs extensive revision. I suggest a major revision.
Authors:
The authors would thank the reviewer for her/his appreciation and indications. The writing and structure of the new version of the paper was extensively revised. Moreover, the new version of the paper was entirely proofread by a professional Proofreading service.
Reviewer1:
- Line 32, use Covid-19, delete Coronavirus Disease 2019, as it as appeared in line 29.
Authors:
In the new version of the manuscript, the authors have done the suggested correction.
Reviewer1:
- Page three is a repetition of page two.
Authors:
The authors have corrected the repetition error.
Reviewer1:
- Line 159, the 2nd citation style appears.
Authors:
The authors have inserted the correct citation style for the reference mentioned.
Reviewer1:
- line 184, "Through Correlations (Pearson’s r) were performed ", the english is problematic.
Authors:
Following the suggestions of the Reviewer, the authors have corrected the English.
Reviewer1:
- Line 209, english is problematic.
Authors:
Following the suggestions of the Reviewer, the authors have corrected the English.
Reviewer1:
- Table 1, the table is hard to read, use * rather than exact p value.
- Table 2, the table is hard to read, use * rather than exact p value.
Authors:
Following the suggestions of the Reviewer, in the new version of Table 1 and Table 2 the exact p value was replaced with *, and the following Note has been added:
“Note. *=p<.050; **=p<.010; ***=p<.001”.
Reviewer1:
- Table 3, the table is hard to read, move the R2, Adj R2 to the below, and make the table cleaner.
Authors:
In the new version of the manuscript, Table 3 has been revised according to the reviewer's indication.

Reviewer 2 Report
The article “Escaping the reality of the pandemic: the role of hopelessness and dissociation in COVID-19 denialism” investigate the sociodemographic and psychological factors associated with COVID-19 denialism expressed by scepticism, nonadherence to guidelines, and negative attitude toward vaccination. This study is well designed and executed, the authors have presented the data in a good manner. The results of this study suggested that hopelessness could exacerbate a defensive dissociative response that could be associated with greater COVID-19 denialism.
I have a few minor comments to improve this manuscript.
Comments
1. In the materials and methods line 140-141, Could the authors be more specific on exclusion and inclusion criteria of participants ? did the authors include vaccinated or non-vaccinated individuals ? if it so please explain the reason for exclusion and inclusion of these and discuss these details in the discussion part of the manuscript.
2. Did the authors consider any specific ethnicity for inclusion or exclusion criteria in the study? Please mention if so.
Author Response
Reviewer 2:
The article “Escaping the reality of the pandemic: the role of hopelessness and dissociation in COVID-19 denialism” investigate the sociodemographic and psychological factors associated with COVID-19 denialism expressed by scepticism, nonadherence to guidelines, and negative attitude toward vaccination. This study is well designed and executed, the authors have presented the data in a good manner. The results of this study suggested that hopelessness could exacerbate a defensive dissociative response that could be associated with greater COVID-19 denialism.
Authors:
The authors thank the Reviewer for her/his appreciation.
Reviewer2:
I have a few minor comments to improve this manuscript.
Comments
- In the materials and methods line 140-141, Could the authors be more specific on exclusion and inclusion criteria of participants ? did the authors include vaccinated or non-vaccinated individuals? if it so please explain the reason for exclusion and inclusion of these and discuss these details in the discussion part of the manuscript.
Authors:
Following the reviewer’s comment, in the new version of the manuscript the exclusion and inclusion criteria of participants were specified more in depth as follows:
“The inclusion criteria were that participants were residents of Italy and that they were able to read and understand Italian. The exclusion criterion was age < 18 years.”.
Moreover, the authors did not distinguish the vaccinated or non-vaccinated individuals and the following sentences have been added in the Discussion section:
“Also, the participants in the present study were not differentiated on the basis of vaccination status. Indeed, the aim of the present study was to highlight factors associated with a negative attitude toward the COVID-19 vaccine and not about how these affected the actual vaccination rate. An interesting future project could investigate whether a greater dissociative response could be found specifically in nonvaccinated individuals.”.
Reviewer2:
- Did the authors consider any specific ethnicity for inclusion or exclusion criteria in the study? Please mention if so.
Authors:
The authors did not consider any specific ethnicity for inclusion or exclusion criteria.

Round 2
Reviewer 1 Report
The authors have addressed all my concerns, I have no comments. I suggest the journal's editoral stuff to check if the length of the paper is appropriate, whether the paper is a little bit short. But anyway, I have no more comments to the content.